# Running Performance Variability among Runners from Different Brazilian States: A Multilevel Approach

**DOI:** 10.3390/ijerph18073781

**Published:** 2021-04-05

**Authors:** Mabliny Thuany, Thayse Natacha Gomes, Lee Hill, Thomas Rosemann, Beat Knechtle, Marcos B. Almeida

**Affiliations:** 1Post-Graduation Program of Physical Education, Federal University of Sergipe (UFS), São Cristóvão, 49100-000 Sergipe, Brazil; mablinysantos@gmail.com (M.T.); thayse_natacha@hotmail.com (T.N.G.); mb.almeida@gmail.com (M.B.A.); 2Division of Gastroenterology & Nutrition, Department of Pediatrics, McMaster University, Hamilton, ON L8N 3Z5, Canada; hilll14@mcmaster.ca; 3Institute of Primary Care, University of Zurich, 8091 Zurich, Switzerland; thomas.rosemann@usz.ch; 4Medbase St. Gallen Am Vadianplatz, Vadianstrasse 26, 9001 St. Gallen, Switzerland

**Keywords:** performance, predictors, multilevel modeling

## Abstract

The ecological model theory highlights that human development (or a given behavior) is the result of the interaction of variables derived from different levels, comprising those directly related to the subjects and those related to the environment. Given that, the purpose of this study is to establish whether runners’ performance may vary among different Brazilian states, as the factors associated with this difference. The sample comprised 1151 Brazilian runners (61.8% men) that completed an online questionnaire, providing information about biological (sex, age, height, and weight), training (running pace, frequency and volume/week, and motivation), sociodemographic (place of residence and wage) aspects, and perceptions about the environmental influences on the practice. Information about state variables was obtained from official institutes, and comprised the human development index (HDI), athletics events, and violence index. Multilevel analysis was conducted in HLM software. State-level characteristics explained ≈3% of the total variance in running performance. Of the total variance explained for the individual level, 56.4% was associated with male sex (β = −54.98; *p* < 0.001), age (β = 1.09; *p* < 0.001), body mass index (β = 6.86; *p* < 0.001), economic status (β = 6.23; *p* = 0.003), the perception of the natural environment (β = 7.58; *p* = 0.02), training frequency (β = −16.64; *p* < 0.001), and weekly volume (β = −0.30; *p* < 0.001). At the state level, only athletics events presented a positive and significant influence on performance. There is a significant role of the environment on the explanation of running performance variability, and given the diversity across states, environmental variables should not be neglected, as they are relevant to the exploration of other variables possibly related to running performance.

## 1. Introduction

Sports performance is a multifactorial trait, determined by different predictors related to both the subject and the environment [1]. Previous studies have analyzed the influence of psychological and physiological aspects [2,3], body composition [4], anthropometric traits [5], and/or environmental characteristics [6], such as training structure [7], birthplace [8], family and coaching support [9], socio-economic, and cultural aspects [10,11], which may influence the level of sporting achievement.

According to ecological model theory, human development (or a given behavior) is the result of the interaction of variables derived from different levels, organized in a hierarchical structure, comprising variables directly related to the subjects as well as those related to social, physical, and natural environments [12,13,14]. In this context, athletes have different life stories, train with different coaches, and live in given neighborhoods, which are located in cities/states/countries with distinct sports policies, different natural environments, and designs [15]. All of these aspects act together to produce the different athletes’ profiles and performance [15,16].

These relationships can be illustrated when considering changes in runners’ performance worldwide during the last decades [17]. Although there has been a significant increase in the number of practitioners across the years [18], improvement in runners’ performance did not follow the same pattern across the continents [19]. For example, South American marathoners showed a drop of around 14% in performance, while their European peers improved by more than 40% [20].

As the largest South American country, Brazil presents nearly continental dimensions and sociocultural contrasts [21], making such differences in runners’ performances notable. There is also discrepancy regarding the distribution of the best Brazilian runners across the states, with a high concentration of them located in the Southeastern region [22]. This result could be related to the existence of local sports policies that promote sport participation among its residents, thereby allowing the development of elite athletes [8,23]. Moreover, each state has different characteristics, such as population size and density; public policies, design, and infrastructure; demographic rates; human development index (HDI) [24]; and violence rates, street safety and security policies [25]; as well as specific geographic and weather characteristics [23,26]. So, it is possible to assume that these particularities can act together to express and explain the differences observed in runners’ performance [27].

Most of the studies conducted so far have primarily focused on understanding the role of individual characteristics on runners’ performance [4,28,29,30]. Therefore, since it is already known that performance is, in part, derived from the inter-relationship between individual and environmental characteristics/contexts, this study aimed to establish whether runners’ performance varies between different Brazilian states, as well as to identify potential factors associated with this performance difference, based on a multilevel modelling approach. Based on previous research that indicated differences according to the best Brazilian athletes [8,31], we hypothesized that nonprofessional runners have a significant difference in performance.

## 2. Materials and Methods

### 2.1. Sample

The sample of the present study was obtained from the InTrack project (https://intrackproject.wixsite.com/website), a study aiming to identify factors associated with road running performance. For the present study, the sample consisted of 1151 runners (61.8% men; 38.2% women), aged between 18 and 72 years old, from 25 states and federal districts comprising the five Brazilian regions (Southeast = 36.3%; North = 7.0%; Northeast = 35.7%; South = 12.4%; and Midwest = 8.2%). To be included in the study, runners were required to answer an online questionnaire. Participants were excluded if they had not answered all the mandatory questions from the applied questionnaire. All participants received information about the study’s purposes and perspectives and gave their written consent to participate.

### 2.2. Instrument

Information obtained from the participants was self-reported, through the questionnaire section “profile characterization and associated factors for runners’ performance”, which was developed and validated previously [32]. The questionnaire provides information in six categories: (1) runner identification (age and sex), (2) anthropometric variables (height and weight), (3) sociodemographic profile (neighborhood, income, educational level, and marital status), (4) perception about the environmental (natural or built) influence on the practice, (5) training variables (volume and frequency/week, sessions/day, practice time, pace (min/km), involvement in official races in the least 12 months, involvement in a running club, the existence of a personal coach to guide the practice, motivation for the practice, and preferred distance), and (6) the family environment (family composition, family members engaged in running practice, involvement in sports during childhood, and family support for sporting involvement during childhood).

The questionnaire was available for eligible participants using an online platform (Google forms), as used in previous studies [33,34,35], between November 2019 and March 2020. This online strategy was chosen to cover all Brazilian states and to maximize the variability between runners. Furthermore, it was not the purpose of the study to obtain a representative sample in each of the Brazilian states, nor nationally.

#### 2.2.1. Individual-level variables

Biological variables

Sex, age, body height, and body weight were self-reported. Body mass index (BMI) was computed by the standard formula: weight (kg)/height (m^2^).

Training variables

Running pace: Running pace was used as the primary performance indicator (included in the model as the outcome variable). Runners were asked to state their run pace in preferred distance.Frequency of training: Runners were asked to state the number of training sessions they complete per week (1–7 train/week). The variable was dichotomized as either at least 3 sessions/week or more than 3 sessions/week.Volume/week: Runners were asked to provide information about the average total distance (in kilometers) they usually cover during their weekly training sessions.

#### 2.2.2. Sociodemographic

Socioeconomic status (SES): Runners were asked to provide an estimate of their monthly income, in a Likert scale format, based on Brazilian minimum wage in 2019 [36]. Answers were restructured in the following categories: low (≤ BRL 998.00 or about < USD 241.06), medium (> BRL 998.00–≤ BRL 2994.00 or about USD 241.06–≤ 723.18), medium–high (> BRL 2994.00–≤ BRL 4990.00 or about > USD 723.18–≤ USD 1205), and high (> BRL 4990.00 or about > USD 1205), which were used in the analysis.Place of residence. Runners were asked about the city they live in (state capital or not).


*Perception of the environmental influences for the practice*


Weather: Runners were asked about their perception of the influence of the natural environment (namely weather conditions) during running practice. Based on their answers, the variable was dichotomized to yes (it influences) or no (it does not influence).Physical structures: The perception about the presence of physical structures and the environment (the existence of parks/places for the practice, street safety and design), that can promote ongoing running practice, was obtained and dichotomized to yes (it influences) or no (it does not influence).

#### 2.2.3. State-level variables

Information was obtained from official institutes, such as the Atlas of Human Development in Brazil [37], the Atlas of Violence [25], and the Brazilian Institute of Geography and Statistics [21] for each state.

Human development index (HDI): Based on the HDI, states were categorized as medium (≤0.699), high (≥0.700 and ≤0.799), or very high (≥0.800) HDI. None of the states had an HDI classified as low (<0.600).Athletics events: Information regarding the existence of athletics events in the various Brazilian states was obtained from State Basic Information Research [38]. The variable was categorized as either yes (there is) or no (there is not).Violence index: Femicide was used as the violence index indicator, obtained from the Atlas of Violence [25]. It expresses the total number of women homicides by year in each state.

### 2.3. Statistical analysis

Descriptive statistics are presented as a mean ± standard deviation (SD) and frequencies, and were computed using IBM SPSS Statistics (IBM Corp. Released 2016. IBM SPSS Statistics for Windows, Version 26.0. Armonk, NY: IBM Corp). The running pace was considered the outcome variable, and the Hierarchical Linear Model (performed in software HLM 6.0) was computed to estimate the association of individual- and state-level variables and performance variance.

A series of hierarchically nested models were fitted, and the model accuracy was analyzed based on the deviance statistics value, which is expected to significantly decrease as the model complexity increases, and this decrease is tested by a chi-square test [39]. After that, the relevance of the predictors included was determined through a pseudo-R^2^ statistical test, which was interpreted as the proportion of the variance reduction for the parameter estimate, which is a result from the comparison of a given model to its previous one [40]. Further, models were built in a stepwise fashion, as generally suggested [39,41]: the first step of the analysis comprised the running of the null model, which allowed computing of the intracluster correlation coefficient to estimate the variance that accounted for states effects on performance. Secondly, the individual-level model (model 1), with the inclusion of the subject predictors (sex; age; BMI; training variables, i.e., frequency and volume/week; sociodemographic factors (SES; place of residence) and individual perception about the influence of the natural and built environment on their practice) was run. Thirdly, the state-level model (model 2) was computed with the insertion of the state-level variables (HDI, violence index, and athletics events). For all the analyses, the significance level was set at *p* < 0.05.

## 3. Results

The sample consisted of runners from both sexes, aged between 18 and 72 years. A total of 61.8% of runners were classified as normal weight according to their BMI. The runners reported an average pace slightly lower than 5:30 min/km, and had a heterogeneous weekly distance covered per week (35.49 ± 29.54 km). The majority of the participants reported practicing for more than one year, and training up to three sessions/week. Runners most frequently lived in state capitals, in states with medium HDI, and with a wide range of feminicide cases per year. Almost 95% of runners received a monthly income above and up to five times the Brazilian minimum wage. Data suggested that it is common the promotion of athletics events in states. Moreover, both the natural and built environment seemed to influence running practice (Table 1).

The final estimated variance at the state level, presented in the null model, was found to be significant, revealing statistical differences across the states. The intracluster correlation coefficient showed that ~3% of the total variance in runners’ performance was explained by the differences between states, meaning that 97% of this variance is explained by runners’ individual characteristics (Table 2).

Figure 1 presents the range of running paces within and between states, sorted by the medians, across Brazilian states. It is possible to observe differences between states, ranging from 252 to 360 s/km, as well as relevant within-state differences.

The model 1 summary shows that women had a pace 54.98 s slower than men (*p* < 0.001); further, with increasing age and BMI, a decrease in the performance was observed (meaning an increase in the time to cover 1 km), i.e., for each year a subject ages, their pace decreases by 1.09 s/km (*p* < 0.001), and for each increase in one BMI unit, there is an increase in 6.86 s in the time to cover one kilometer (*p* < 0.001). Moreover, runners with more than three training units/week, and those who cover more km/week in training sessions, tended to have a faster pace (a pace decrease of 16.64 s if subjects train more than 3 sessions/week, an increase in performance of about 0.30 s/km for each additional kilometer in weekly volume; *p* < 0.001). The higher the SES category, the worse the running performance, with an increase in pace of about 6.23 s (*p* = 0.003). Similarly, runners who perceive the climate as a factor that influences their practice showed a pace of about 7.58 s (*p* = 0.02) slower than their peers that do not have this perception. No statistically significant associations were found between the place where runners live (capital or not capital) or their perception about the influence of physical structure on their running practice (*p* > 0.05). The model 1 deviance was significantly lower than the null model’s deviance (Δ = −2501.67220, *p* <0.001); given this, it was possible to estimate the proportion of runners’ variables on the performance variance, where 54.6% of the within-state variance was attributed to individual characteristics inserted in the model, while these same variables explained 56.5% of the between-state variance.

Model 2 investigated the states effect on the performance variance. From the set of the variables included in the model, runners who lived in states with athletic events had a faster pace (−9.30 s, *p* = 0.001); while no statistically significant associations were observed for HDI or violence index and runners’ performance (*p* > 0.05). Results regarding the individual-level variables remained similar. The difference between model 2 and model 1 deviance was significant (Δ = −8.48100, *p* = 0.036), and the estimation of the proportion of states’ characteristics in the explanation of the variance in the performance showed that about 90.6% of the between-state variance was explained by the model built.

## 4. Discussion

The purpose of this study was to establish whether runners’ performance varies between different states, as well as to identify the factors associated with this performance difference, based on a multilevel modelling approach. There was relevant between- and within-state variance in running performance that was somewhat expected, since previous studies have already reported discrepancies regarding the distribution of the best runners [19,42], or even distribution of Brazilian elite athletes [8,23] across states, meaning that differences in the pace of non-elite runners between states would be expected. Moreover, even within states, there is a diversity of social, cultural, and economic aspects, which, in association with runners’ motivation/interests, would probably influence and differentiate their performance. In line with previous studies, the present results showed that running performance results from the interaction between biological, sociodemographic, and training variables.

Biological variables included in the model (sex, age, and BMI) were associated with running performance in previous studies [29,43,44]. This sex dimorphism, which has been largely reported and investigated, is usually associated with several physiological and cultural characteristics that seem to favor men [45]. Although evidence suggests that sex differences in sports performance stem from cultural/social factors [45,46], women tended to underperform in most of the sport modalities. In the present study, women took an average of 55 s longer to cover 1 km when compared with men. At a marathon-like distance, this difference adds more than 38 min to finish the race [47].

Age and BMI values corroborate preceding evidence that highlights an inverse relationship between age, BMI, and performance [17,43], as older runners and those with higher BMI had a slower pace. BMI is strongly related to the power required to transpose a given body weight [43]. This is a key component of energy expenditure in running, making heavier subjects spend more energy running [43]. Furthermore, since running is a sensitive weight sport [48], a 10% boost in an athlete’s body weight is associated with an approximately 14% increase in energy cost [49]. Additional body weight could represent a critical mechanical load rise in the leg swing movement. Therefore, more energy is spent to accelerate the center of mass [43], compromising running economy [50].

Although the age effect on performance seems to be small in the present study (1.09 s per year of age), a study conducted by Nikolaidis et al. [51] identified that the peak performance in running occurs approximately between ages 35 and 39 in women, and under the age of 35 years in men. Behavioral and physiological changes, such as a reduction in daily physical activity and training habits, and a decline in cardiorespiratory fitness, may explain this fact. It is known that changes in biomechanical and muscular function should also lead to changes in performance, and these aspects tend to decline with increasing age [4], leading to a decrease in performance.

It is well-known that the manipulation of training variables can increase performance given their effect on both physical fitness and physiological capabilities [52]. In the present study, training frequency and volume were included in the analysis because they are frequently reported as training behavior [53], and were significantly associated with the runners’ performance (*p* < 0.001). Noticeably, training exposure may lead to significant performance increases due to relevant changes in maximal oxygen uptake (VO_2_max), lactate threshold, and running economy [54].

As road running is carried out in natural environments, training (and performance, as a consequence) may be influenced by weather aspects, such as temperature, relative humidity, and wind [55]. Based on geographical differences between Brazilian states [56], runners’ perceptions about the role of environment on running practice was included as a possible predictor of the performance variance. The results showed runners who believed that natural environment plays a relevant role on practice demonstrated worse performance. Previous studies demonstrated that thermal sensation and discomfort are associated with changes in body temperature, which can lead to decreases in performance when it reaches and/or exceed 39 °C [49]. Nevertheless, in the present study, runners were not asked whether body temperature changes could take them out of their comfort zone and cause a negative impact on performance. In addition, running is an activity that can be performed in indoor environments (such on a treadmill), which can attenuate problems related to natural environmental constraints. However, this option is not always accessible for all runners, given that it involves access to places where they may have to pay or require a membership.

The relationship between economic conditions and sports practice is a controversial point that has been debated for years. Although road running is likely accessible for most people, it is unquestionable that the use of more sophisticated equipment is not available for all. Hence, subjects with better economic conditions have facilitated access to sports equipment/gadgets and are more prone to continue their running practice [57]. At the international level, socio-economic rise is one of the flagships for involvement in sports practice [11], meaning that low-income athletes aspire to achieve better economic conditions through sport participation. This scenario may explain why performance was inversely related to runners’ average income in the present study, with runners from the lowest economic classes presenting better average running pace (285.18 s/km) than medium (317.45 s/km), medium–high (335.36 s/km), and high (355.00 s/km) classes.

Some studies have focused on understanding the role of social and built environmental characteristics, such as street safety, neighborhood socio-economic status, access to sports facilities, urban density and design, and an attractive environment, and their relationship with sports participation [58,59,60]. However, the present results revealed that only the existence of athletic events in the state showed a significant relationship with running performance, in that runners living in these states showed a faster pace (about 9 s/km faster). These states probably investment more in sport infrastructure and have better sports policies, which can motivate runners to train and improve performance [61].

Despite the understanding that feminicide and HDI indicate, respectively, safety and socio-economic conditions that can influence training attendance [58], the statistical model revealed that these variables did not affect running performance variance. Furthermore, states from the South and Southeast regions are those considered to have friendliest environment for physical activity [62], and this scenario may promote the involvement of residents in running practice, leading to improved performance. It is interesting to note that those states are also the ones over-represented in rankings of the best Brazilian runners [42]. These regions also concentrate most of the Brazilian elite athletes, display a more accessible environment to sport practice, have a traditional sport culture [8,23], and are also those with the highest HDIs in Brazil [37]. In our results, these regions also possessed the best average pace, which indicates better performance (South = 310.5 ± 52.0 s/km; Southeast = 322.6 ± 56.4 s/km). Further studies should address other characteristics, such as political preferences, religious/cultural systems, and sociodemographic aspects. Additionally, sports organizations seem to play a relevant role in facilitating athletic training, and their role must not be omitted.

The reduction in the variance at the state level in the final model observed in this study was reasonably expected based on several reasons. The inclusion of individual variables in model 1 would lead to a reduction in the within-state variance, thus also leading to a reduction in between-state variance. When analyzing the systems structures based on the hierarchical and bioecological models, the state environment is located some distance from the microsystem, so a lesser influence of this system level on the runners’ performance would be expected. However, there is difficulty in obtaining information at the state level that can directly influence running performance. This is largely due to the limited available research and the inaccessibility of state-level information.

Despite all efforts, this study is not free of limitations, which may restrain generalization of results. There was a discrepancy in sample distribution across Brazilian states. However, multilevel analysis is robust enough to avoid bias in the main analysis due to this fact. Self-reported information could be susceptible to misleading data. Despite this, other studies have successfully used this strategy. Finally, we faced some difficulty obtaining information related to states’ aspects/characteristics that could be used as predictors in the second-level model. Future researches should consider exploring other variables related to participants (e.g., physical fitness, body composition, somatotype, nutritional aspects) and states (e.g., events and running clubs (number and cost), and temperature). Therefore, to the best of our knowledge, this is the first country-based study to map and analyze running predictors based on a bioecological approach, including variables from the macrosystem level.

Given the relevant diversity across states’ environments, it seems relevant to explore the wide range of other possible variables related to running performance. In addition, it is important to note that information regarding the role of variables that can be changed with training in the expression of the performance can be used by coaches and runners aiming to improve their performance. Moreover, public policies should be developed focusing on providing changes in urban design and security, favoring the practice of physical activities by the population (such as running) and more accessible sport events.

## 5. Conclusions

State-level differences explained 3% of the total variance in Brazilian runners’ performance, with 54.4% of this variance being explained by running events, economic, and safety aspects. At the individual level, biological (sex, age, and BMI), sociodemographic (SES), and training (training frequency and volume/week) variables explained 56.4% of the 97% of the variance fraction associated with individual-level characteristics.

## Figures and Tables

**Figure 1 ijerph-18-03781-f001:**
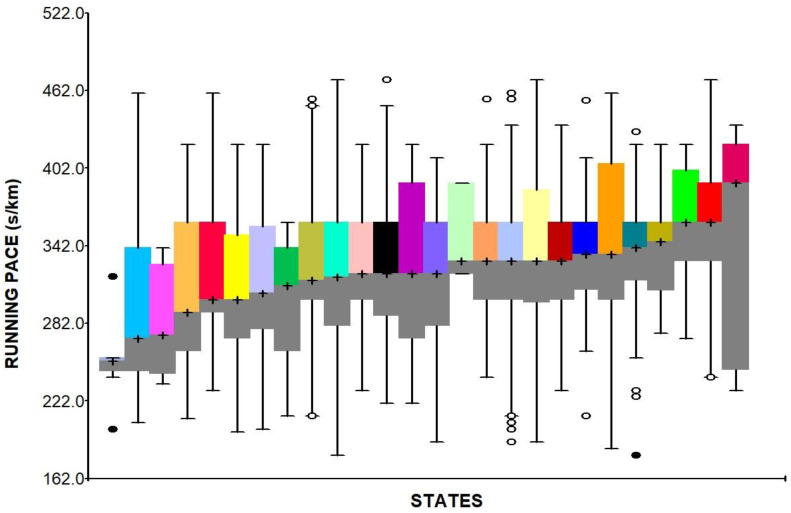
Running pace variance between Brazilian states, sorted by the median. Box-plot elements are as follow: cross, median; superior and inferior box limits, mean 75th and 25th percentiles, respectively; top and bottom bars, maximum and minimum normal values, respectively; circles indicate outliers.

**Table 1 ijerph-18-03781-t001:** Descriptive statistics (means and standard deviation; or frequency) of the individual- and state-level variables.

Variables	Mean (Standard Deviation) or Frequency (%)
*Sex*	
Male	711 (61.8%)
Female	440 (38.2%)
*Age (years)*	37.9 (9.4)
*BMI (kg·m* ^−2^ *)*	24.3 (3.1)
*Practice time*	
≤1 year	173 (15.0%)
>1 year	976 (84.8%)
*Running pace (s)*	324.2 (57.7)
*Volume training/week (km)*	35.5 (29.5)
*Frequency training/week*	
≤3 sessions/week	678 (58.9%)
>3 sessions/week	473 (41.1%)
*Live in capital*	
No	512 (44.5%)
Yes	639 (55.5%)
*Socioeconomic status (SES)*	
Low	65 (5.6%)
Medium	542 (47.1%)
Medium-high	526 (45.7%)
High	4 (0.3%)
*Natural environment influences*	
No	371 (32.2%)
Yes	779 (67.7%)
*Physical environment influences*	
No	299 (26.0%)
Yes	852 (74.0%)
*Athletics events*	
No	5 (19.2%)
Yes	21 (80.8%)
*Human development index*	
Medium	13 (50.0%)
High	12 (46.2%)
Very High	1 (3.8%)
*Female homicides*	187.4 (150.1)

**Table 2 ijerph-18-03781-t002:** Summary of results of the hierarchical linear model for the variance in running performance.

Parameters	Null Model			Model 1			Model 2		
	Estimates	Standard Error	*p*-Value	Estimates	Standard Error	*p*-Value	Estimates	Standard Error	*p*-Value
Intercept	323.65	2.92	<0.001	346.49	5.81	<0.001	356.81	5.36	<0.001
Sex				−54.98	3.08	<0.001	−55.25	3.13	<0.001
Age				1.09	0.12	<0.001	1.12	0.12	<0.001
BMI				6.86	0.52	<0.001	6.88	0.52	<0.001
Place of residence				0.02	3.17	0.995	−0.15	3.30	0.963
SES				6.23	2.06	0.003	6.42	2.01	0.002
Natural environment				7.58	3.25	0.02	7.50	3.26	0.022
Physical structure				3.89	2.71	0.152	3.96	2.68	0.140
Frequency/week				−16.64	2.65	<0.001	−16.45	2.55	<0.001
Volume/week				−0.30	0.08	<0.001	−0.30	0.08	<0.001
Athletic events							−9.36	2.34	0.001
Woman homicides							−0.01	0.01	0.139
HDI							−5.21	2.93	0.089
Variance components: random effects									
Between-states	104.13			45.34			9.79		
Within-sates	3261.73			1479.73			1484.38		
Model summary									
Deviance statistic		12,595.210			10,093.54			100,85.06	
Number of estimated parameters		3			12			15	

BMI, Body Mass Index; SES, Socioeconomic status; HDI, Human Development Index.

## Data Availability

The data are not publicly available due to ethical concerns.

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
