# Peer review of "Running Performance Variability among Runners from Different Brazilian States: A Multilevel Approach"

_ijerph, 2021, doi:10.3390/ijerph18073781_

Round 1

Reviewer 1 Report

I commend the authors on the completion of this manuscript. Overall it is well written and on an important topic. I have a few concerns highlighted below.

Line 27: Abstract. “Considering the environment on the expression of the performance, variables from this level should not be neglected, and strategies to provide a more secure and “friendly” environment for the practice should be developed”. I think it must be better to reflect exactly the conclusion extracted from the study data. This sentence is too ambiguous and in some sense remote from the results of the study.

Line152. “Descriptive statistics are presented as mean ± standard deviation (SD) and frequencies and was computed”.  It must be “…were computed.”

Line 198. “Model 1 summary shows that women have a pace 54.98 sec higher than men (p<0.001); further, with increasing age (1.09 s, p<0.001) and increasing BMI (6.86 s, p<0.001), a decrease in the performance was observed (meaning increasing the time to cover 1 km).” I think that increasing units must be explained: Age: 1.09 s per km, per year?  BMI: 6.86 s? I think this is  an error.

Line 346: Conclusions: “At the individual-level, biological, sociodemographic, and training variables explained 56.4% of the 97% of the variance fraction associated with the individual-level.” I think it must be better to determine what biological, sociodemographic, and training variables concretely explained performance. For example, biological: sex and age.

Line 348. I Think the next paragraph must be included in the discussion section, as study prospective and clinical implications: “Given that there is relevant diversity across states environments, it seems relevant to explore the wide range of other possible variables related to running performance. In addition, it is important to note that information regarding the role of variables that can be changed with training in the expression of performance can be used by coaches and runners aiming to improve their performance. Moreover, public policies should be developed focusing on providing changes in urban design and security, favoring the practice of physical activities by the population (such as the running practice) and the more accessible sport events.”

Author Response

Comments and Suggestions for Authors

I commend the authors on the completion of this manuscript. Overall it is well written and on an important topic. I have a few concerns highlighted below.

Line 27: Abstract. “Considering the environment on the expression of the performance, variables from this level should not be neglected, and strategies to provide a more secure and “friendly” environment for the practice should be developed”. I think it must be better to reflect exactly the conclusion extracted from the study data. This sentence is too ambiguous and in some sense remote from the results of the study.

Authors’ answer: We performed as suggested, re-written the sentence.

Line152. “Descriptive statistics are presented as mean ± standard deviation (SD) and frequencies and was computed”.  It must be “…were computed.”

Authors’ answer: Changed.

 Line 198. “Model 1 summary shows that women have a pace 54.98 sec higher than men (p<0.001); further, with increasing age (1.09 s, p<0.001) and increasing BMI (6.86 s, p<0.001), a decrease in the performance was observed (meaning increasing the time to cover 1 km).” I think that increasing units must be explained: Age: 1.09 s per km, per year?  BMI: 6.86 s? I think this is  an error.

Authors’ answer: Sentence was changed, and we tried to better plain the results.

Line 346: Conclusions: “At the individual-level, biological, sociodemographic, and training variables explained 56.4% of the 97% of the variance fraction associated with the individual-level.” I think it must be better to determine what biological, sociodemographic, and training variables concretely explained performance. For example, biological: sex and age.

Authors’ answer: Changed

Line 348. I Think the next paragraph must be included in the discussion section, as study prospective and clinical implications: “Given that there is relevant diversity across states environments, it seems relevant to explore the wide range of other possible variables related to running performance. In addition, it is important to note that information regarding the role of variables that can be changed with training in the expression of performance can be used by coaches and runners aiming to improve their performance. Moreover, public policies should be developed focusing on providing changes in urban design and security, favoring the practice of physical activities by the population (such as the running practice) and the more accessible sport events.”

Authors’ answer: We appreciate the suggestion, and the paragraph was included in the end of the discussion section.

Reviewer 2 Report

Thank for the opportunity to review this paper. The study is really well written and the data analysis is intriguing, but I must necessarily underline and suggest greater clarity in the explanation of the method used. Hierarchical Linear Modeling is a difficult approach for the reader and therefore must be studied in depth in the methods to make the results clearer.

Abstract: I recommend providing a background for the purpose of the study as in the introduction, so I would suggest a nod to the concept of "ecological model theory".

Methods:

148: I do not understand the correlation between the Violence index and the femicide rate. But above all, how does the information on how runners' performances change with the femicide rate?

158 A series of hierarchically nested models were fitted, and the model accuracy was analyzed based on the Deviance statistics value, where it is expected to occur a significant decrease in it as the model complexity increases.
I do not find the method of analysis clear, perhaps enriching it and above all by inserting at least 2 bibliographical references I could legitimize it.

198-222. I would suggest inserting a complete HLM pathway into the methods, so as to put into the results, what has been achieved by studying hierarchical models

Author Response

Comments and Suggestions for Authors

Thank for the opportunity to review this paper. The study is really well written and the data analysis is intriguing, but I must necessarily underline and suggest greater clarity in the explanation of the method used. Hierarchical Linear Modeling is a difficult approach for the reader and therefore must be studied in depth in the methods to make the results clearer.

Authors’ answer: We thanks reviewer for comments. We tried to present more details, and also references, in the statistical procedures.

Abstract: I recommend providing a background for the purpose of the study as in the introduction, so I would suggest a nod to the concept of "ecological model theory".

Authors’ answer: Information inserted.

Methods:

148: I do not understand the correlation between the Violence index and the femicide rate. But above all, how does the information on how runners' performances change with the femicide rate?

Authors’ answer: We thanks the reviewer for the opportunity to clarify this point. Please, note that notwithstanding feminicide does not be considered as “the” violence index of a given population, it represents one part of this index, and for this reason we used it as a violence index in the present study. In addition, it is important to note that places with higher feminicide index are seem as not friendly for women, given that it means that there is a high probability of women be killed/murdered (just because they are women), meaning that they do not feel safe. Since running is a practice performed, usually, in outdoor spaces, in cities where the feminicide index is high, probably the general violence will be affected by this fact, but more important, this highlights that women may not feel confident/comfortable to perform outdoor activities (such as running) alone, and this can reflect in their performance (lowest training frequency and/or volume, for example). At the introduction, we presented references that show the relationship between “violence and running practice”.   

158 A series of hierarchically nested models were fitted, and the model accuracy was analyzed based on the Deviance statistics value, where it is expected to occur a significant decrease in it as the model complexity increases.
I do not find the method of analysis clear, perhaps enriching it and above all by inserting at least 2 bibliographical references I could legitimize it.

Authors’ answer: The description of statistical analysis was improved as suggested.

198-222. I would suggest inserting a complete HLM pathway into the methods, so as to put into the results, what has been achieved by studying hierarchical models.

Authors’ answer: We included more information at methods section, regarding statistical analysis/procedures, and also tried to present more information at results section.

Round 2

Reviewer 2 Report

Thanks for the further review. I feel the manuscript has improved.